# Empowering Large Language Model Agents through Action Learning

**Haiteng Zhao**[1]   **Chang Ma**[3]   **Guoyin Wang**[2]   **Jing Su**[2]   **Lingpeng Kong**[3]
**Jingjing Xu**[2]   **Zhi-Hong Deng**[1]   **Hongxia Yang**[2]
[1] Peking University [2] ByteDance [3] The University of Hong Kong
{zhaohaiteng, zhdeng}@pku.edu.cn        {cma, lpk}@cs.hku.hk
guoyinwang.duke@gmail.com  {sujing.29, xujingjing, hx.yang}@bytedance.com

## Abstract

Large Language Model (LLM) Agents have recently garnered increasing interest yet they are limited in their ability to learn from trial and error, a key element of intelligent behavior. In this work, we argue that the capacity to learn new actions from experience is fundamental to the advancement of learning in LLM agents. While humans naturally expand their action spaces and develop skills through experiential learning, LLM agents typically operate within fixed action spaces, limiting their potential for growth. To address these challenges, our study explores open-action learning for language agents. We introduce a framework LearnAct with an iterative learning strategy to create and improve actions in the form of Python functions. In each iteration, LLM revises and updates the currently available actions based on the errors identified in unsuccessful training tasks, thereby enhancing action effectiveness. Our experimental evaluations across Robotic Planning and Alfworld environments reveal that after learning on a few training task instances, our approach to open-action learning markedly improves agent performance for the type of task—by 32% in AlfWorld compared to ReAct+Reflexion, for instance— highlighting the importance of experiential action learning in the development of more intelligent LLM agents.

## 1 Introduction

Language agents, which employ the large language model (LLM) as the policy model to control agents to iteratively take actions and interact with environments, have recently garnered increasing interest (Yao et al., 2023a; Brohan et al., 2023; Wang et al., 2023a; Xie et al., 2023; Song et al., 2023; Xu et al., 2023). The underlying reason is that LLMs offer a fresh angle to address the *commonsense* issue that is difficult to tackle in the reinforcement learning paradigm which learns the agent policy solely from trial and error. Reasoning and planning of agents are often derived from prior knowledge in the particular environment, precisely where LLMs excel.

Researchers increasingly recognize that, while leveraging LLMs for agent control shows promise, it is far from perfect due to their limited ability to learn from experience (Yao et al., 2023b; Shinn et al., 2023; Huang et al., 2023). The substantial scale of LLMs makes direct policy model finetuning impractical. Instead, LLMs depend on incorporating historical interactions into prompts to leverage past experiences for future action planning (Yao et al., 2023a; Shinn et al., 2023; Sun et al., 2023; Madaan et al., 2023). However, these approaches are often constrained in their capacity to learn from long-term experiences and typically draw experience on single-instance (Wang et al., 2023c; Ma et al., 2024; Wang et al., 2023a).

While humans naturally expand their action spaces and develop skills through experiential learning, LLM agents typically operate within predetermined action spaces (Qin et al., 2023; Schick et al., 2023), limiting their potential for growth. In this work, we propose a novel learning paradigm for LLM agents that focuses on learning to expand and iteratively refine

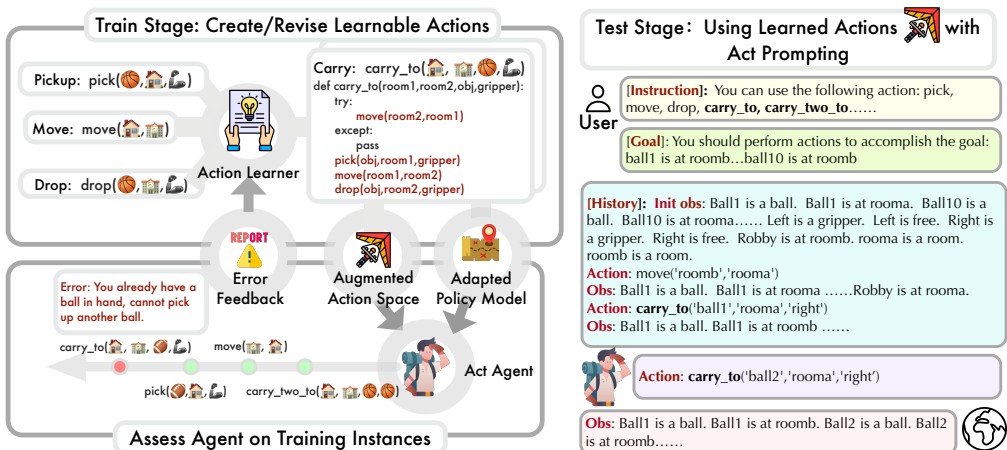

Figure 1: Illustration of the training and test stage of LearnAct: **Left**: During the training stage, LearnAct expands the action space by first creating actions and then optimizing them based on the execution feedback. **Right**: The test stage uses learned action space to facilitate sequential decision-making. The prompting format follows the Act (Yao et al., 2023a) agent.

the action space, thereby aligning tasks more closely with the agents' planning abilities. By adapting the action space to fit the LLM's planning, we address the limitations imposed by fixed action spaces, such as the misalignment between commonsense knowledge-guided planning and actions, and the prevalence of action errors due to unmet prerequisites or ineffective strategies(Gu et al., 2022; Ma et al., 2024; Ahn et al., 2022). This approach not only mitigates bottlenecks in language agent performance but also allows transferring experience across different tasks.

For a more illustrative view, we could look at the example where an LLM agent is asked to make cocktails. With a learnable and open action space, LLMs may naturally instruct the agent to gather all the components in a specific order and mix them, while the closed action space may only involve basic handling, such as moving to different places and grabbing bottles, which greatly increases the difficulty of the task for LLM agent. On the other hand, it is necessary to update existing actions to accommodate changing factors in the environment. Still, in the cocktail example, when making an *Old Fashioned* cocktail, muddling sugar is often the standard practice. However, if only syrup is available, the agent should adapt its pre-set actions to prevent repeated failures despite being competent to accomplish the task.

To solve these issues, we conduct an extensive exploration of action learning in LLM agents. We introduce a framework LearnAct, designed to generate and optimize new action types as APIs dynamically. The newly generated action types are in the form of Python functions, leveraging the LLM's extensive prior knowledge and code generation ability to devise a diverse and representative action space. Furthermore, LearnAct stands out due to its iterative learning strategy, which continuously refines actions through a feedback loop. In each cycle, the LLM evaluates the effectiveness of current actions using training examples, identifying and rectifying errors in failed instances. The learning strategy progressively deepens task understanding and improves learnable actions.

Our experimental results demonstrate that this method of iterative refinement not only creates complex and user-friendly action types but also achieves more effective and efficient learning compared to previous state-of-the-art methods such as Reflexion (Shinn et al., 2023). By learning on a few problem instances, LearnAct can generalize to a general type of task with strong performance. In summary, our contributions are as follows.

- We propose the action learning framework for interactive decision making language model agents by generating and updating learnable actions, enabling learning customized action space that better fits the LLM's planning capacity.

- To implement the action learning framework, our method LearnAct employs Python functions to generate new actions, enabling flexible definitions of action types. Our iterative learning strategy incorporates LLM to autonomously refine

and updates the currently available actions based on errors in failed tasks.

- Our experimental results demonstrate that LearnAct effectively learns action spaces within a few trials, acquiring transferable capabilities in diverse environments. Through action learning, LearnAct outperforms SOTA agents by a significant margin, showcasing the potential of action learning for LLM agents.

## 2 Related Work

**LLM Agent Learning** Recent research has advanced the use of large language models (LLMs) in embodied agents (Duan et al., 2022; Huang et al., 2022a; Ahn et al., 2022; Huang et al., 2022b; Yao et al., 2023a; Park et al., 2023; Wu et al., 2023; Sun et al., 2023; Ma et al., 2024; Yuan et al., 2023; Qiao et al., 2023; Stengel-Eskin et al., 2024). These methods differ from previous reasoning-centered methods like Chain-of-Thought (Wei et al., 2022; Chen et al., 2022; Liang et al., 2023; Wang et al., 2023b) as they iteratively incorporate environment feedback to modify subsequent plans. This *closed-loop* process is centered on the ability of LLM agents to learn from environment.

The majority of previous work on multi-turn reflection concentrates on learning from environment feedback within the *same* problem (Lai et al., 2023; Le et al., 2022; Chen et al., 2023; Liu & Abbeel, 2023; Singh et al., 2023). ReAct (Yao et al., 2023a) propose a basic framework for using thought to enforce the LLM reflect on previous behaviors within the same trial. Reflexion (Shinn et al., 2023; Park et al., 2023), on the other hand, uses a multi-trial approach to perform reflection. By summarizing historical experiences and identifying reasons for failure, LLM is prompted to generate insights for more effective agent instruction. Similarly, ExpeL (Zhao et al., 2023) facilitates a general learning system that extracts insights and past experiences with retrieval. Retroformer (Yao et al., 2023b) propels introduce prompt-based training to enhance LLM reflections.

Our method differs from this line of work in two points (1) LearnAct performs direct learning in the action space, which substantially improves the reliability and utility of generated actions. (2) LearnAct does not target a single problem instance but learns experience from a few training instances and is tested on a general type of task. A more detailed comparison is provided in Table 1.

**Hierarchical Reinforcement Learning** LearnAct generates new action types based on atomic actions to enable more informative and applicable actions, similar to the approach in hierarchical reinforcement learning (Erol et al., 1994; Kulkarni et al., 2016; Bacon et al., 2017), where a high-level executor plan with high-level action types, and then executes the seed actions according to the high-level plan (Yao et al., 2023a; Sharma et al., 2022; Wang et al., 2023a; Sun et al., 2023). Our approach also differs from hierarchical reinforcement learning by not only using code-based action space, providing stronger capacity due to the code statements such as conditions and loops (Cai et al., 2023; Qian et al., 2023), but also avoiding the typical imposition of a strict two-level structure, using flexible mixed-level actions instead.

| | Closed Loop | Action Type | Learnable | Transferable | Learnable Module |
|---|---|---|---|---|---|
| ReAct | ✔ | Natural language | ✗ | - | - |
| ReAct+Reflexion | ✔ | Natural language | ✔ | ✗ | Policy |
| CodeAsPolicy | ✗ | Code | ✗ | - | - |
| Voyager | ✔ | Code | ✔ | ✔ | Action |
| LearnAct | ✔ | Code | ✔ | ✔ | Action |

Table 1: The comparison of our method and other open action agents. ✔Voyager used a curriculum learning pipeline tailored for Minedojo to learn new actions.

## 3 Problem Statement

The task of a language agent can be succinctly modeled as a Partially Observable Markov Decision Process (POMDP), which is defined by a tuple $\langle \mathcal{S}, \mathcal{O}, \mathcal{A}, \mathcal{T}, \mathcal{R} \rangle$, with $\mathcal{S}$ representing

---

**Algorithm 1** Training Process of LearnAct

---

    **Input:** Training Set $\mathbb{D}_{train} = \{\mathcal{E}_1, \ldots, \mathcal{E}_M\}$, Language Agent $G$, Learner LLM $F$.
    **Input:** Original Actions $\mathcal{A}_0$, Basic Task Instruction $\pi_0$
    # *Assess action space*
    **function SolveProblem**$(\pi, \mathcal{E}, \mathcal{A}, G)$
        **for** $t = 1$ **to** maxsteps **do**
            $a_t = G(\pi, \mathcal{E}, h_t)$
            Execute $a_t$, add observation or fail info to $h_t$
        **end for**
        **return** IsSolved$(\mathcal{E}, h_t)$
    **end function**
    # *Training Procedure*
    $\mathcal{A}_1, \pi_{\mathcal{A}_1} = $ ActionCreation$(\pi_0, F)$
    **for** $i = 1$ **to** maxiter **do**
        results = SolveProblem$(\pi_0 + \pi_{\mathcal{A}_i}, \mathcal{E}_m, \mathcal{A}_0 \cup \mathcal{A}_i, G), m = 1$ **to** $M$
        $s_{1:T}, a_{1:T}, r = $ SelectErrorCase$(results)$
        $\mathcal{A}_{i+1}, \pi_{\mathcal{A}_{i+1}} = $ ActionLearn$(\mathcal{A}_i, s_{1:T}, a_{1:T}, r, F)$
    **end for**
    **Return** $A_{\text{last}}, \pi_{A_{\text{last}}}$

---

the set of all possible states, $\mathcal{O}$ being the observation space through which the agent perceives the state, $\mathcal{A}$ denoting the action space, $\mathcal{T} : \mathcal{S} \times \mathcal{A} \rightarrow \mathcal{S}$ being the state transition function, and $\mathcal{R} : \mathcal{S} \times \mathcal{A} \rightarrow \{0, 1\}$ being the reward function.

At step $t$, the language agent $G$ takes an action step $a_t$ based on policy $\pi$ (policy prompt), problem $\mathcal{E}$ (problem instructions), and previous observation-action trajectory $h_t = [o_1, a_1, \ldots, o_{t-1}, a_{t-1}, o_t]$,

$$a_t = G(\pi, \mathcal{E}, h_t) \tag{1}$$

$\pi$, $\mathcal{E}$, and $h_t$ correspond to the instruction, goal, and history in Figure 1 (right), respectively. Action $a_t$ is successfully grounded (executable) only if it is within the action space, i.e. $a_t \in \mathcal{A}$. Even successfully grounded action may be invalid due to unmet conditions or has no effect in the current state. The agent aims to maximize the final reward $r$. In this work, we use an outcome-based reward mechanism (ORM) (Uesato et al., 2022) that assigns a binary indicator as a reward, depending on whether the task has been successfully completed.

In this work, we try to answer the question *How to learn from experience and apply them to other decision-making problems?* We focus on the training-testing setting, where for each task, our LLM agent optimizes its action space and corresponding policy on the training set $\mathbb{D}_{\text{train}} = \{\mathcal{E}_1, \ldots, \mathcal{E}_M\}$, before trying to accomplish problems in the test set $\mathbb{D}_{\text{test}} = \{\mathcal{E}_{M+1}, \ldots, \mathcal{E}_{M+N}\}$. These problems share the same original action spaces and rules, though requiring varied policies to solve them as they have different scenarios and goals. This setting poses a challenge for the agent to develop a general capacity through learning to accumulate experience and accomplish *tasks of this general type*.

To address this problem, we introduce an action-learning framework designed for language agents, enabling them to autonomously enhance their skills by learning from interactions within their environment, as shown in Figure 1. The framework considers the action space, denoted as $\mathcal{A}$, to be an open set capable of learning and expanding. More specifically, we define the original action space as $\mathcal{A}_0$ and augment it with newly learned action type (API) $\mathcal{A}'$, represented as $\mathcal{A}_0 \cup \mathcal{A}'$. Also, instruction for using these new actions $\pi_{\mathcal{A}'}$ is updated to the original policy instructions $\pi_0$, i.e. the new policy is denoted as $\pi_0 + \pi_{\mathcal{A}'}$. Consequently, subsequent actions by this LLM agent are characterized as $a_t = G(\pi_0 + \pi_{\mathcal{A}'}, \mathcal{E}, h_t), a_t \in \mathcal{A}_0 \cup \mathcal{A}'$, as depicted in Figure 1 (right), where the agent is guided to generate both origin and learned actions.

# 4 Method

The overall pipeline of LearnAct is illustrated in Figure 1 and Algorithm 1. The training stage of LearnAct involves first creating new actions and then refine them based on the error feedback on training samples. After learning the action space and policy instructions in the training stage, the refined agent perform actions from the augmented action space and try to accomplish the problem step by step. The training stage is specified in §4.1, and test stage is detailed in §4.2.

## 4.1 Training Stage

**1. Expanding Action Space by Action Creation.** Our method first creates custom action types, akin to API, to enable LLMs to interact with the environment more seamlessly, denoted as ActionCreation. These newly crafted action types are implemented as code functions, thereby unlocking the potential for more intricate logical expressions that leverage the foundational APIs through constructs such as conditional statements (if-else), loops, and assertions. ActionCreation involves two steps:

(1) *Generating Action Function $\mathcal{A}'$*: Upon receiving detailed instructions and problem specifics, the LLM, denoted as $F$, is prompted to summarize high-level actions to complete this task in the form of Python functions for agent use. The detailed prompt is in the Appendix. These functions can call multiple basic or defined actions to complete subtasks. After generation, functions are parsed and added to the action space. See detailed prompt in Appendix.

(2) *Generating Policy Instruction $\pi_{\mathcal{A}'}$ for Using New Action*: After generating new action functions, we then update the agent with information about their potential applications. As the basic instruction contains action descriptions and usage examples, the updated policy guidance includes both a description of each function and a usage example.

First, the LLM $F$ is prompted to provide a comprehensive description of each new function. The purpose of this description is to offer the agent an overview of the anticipated outcomes and necessary conditions, along with the input-output format of the new action. Second, the language model generates illustrative usage examples for the new functions. This is crucial for guiding LLM to use the new action in the appropriate scenario, as indicated by previous work (Schick et al., 2023). Prompts are in Appendix.

We denote this process by the `ActionCreation`:

$$\mathcal{A}_1, \pi_{\mathcal{A}_1} = \texttt{ActionCreation}(\pi_0, F), \tag{2}$$

**2. Learning Actions Based on Error Feedback** After creating the initial set of actions, there's a possibility that these actions may have errors, misunderstand the task, overlook certain task scenarios, or possibility for misuse. To address this, we devise the training phase that allows the language model to learn through trial and error, as shown in Figure 2. In this phase, the agent is executed in the training instances, identifies action failures, and then iteratively updates the action set, until successfully solving all the instances with no action errors or exceeding maximum optimization steps.

During the learning process, the agent first tries to solve problems in the training set with the current available action space $\mathcal{A}$ and policy instruction $\pi_{\mathcal{A}}$, denoted as the `SolveProblem` process. Then a failed problem with action error is sampled with `SelectErrorCase` operator and then the `ActionLearn` either fixes it by revising the used action or writing note as an annotation. We present each of them below:

(1) `SolveProblem`: The assessment of current policy and action space is done by trying to solve problems in the training set. We follow the simple Act(Yao et al., 2023a) agent and prompt the agent to solve the task by interacting with the environment and generating actions based on history.

(2) `SelectErrorCase`: This step identifies error-inducing steps in action sequences based on the environment's feedback, such as invalid actions due to unmet preconditions, ineffective actions, or errors in action names or parameters. For learnable actions composed of multiple atomic steps, it evaluates errors in each atomic step execution.

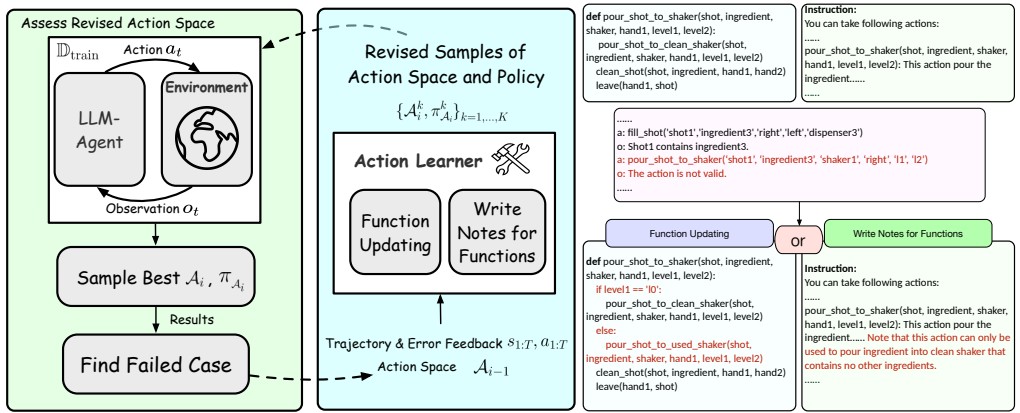

Figure 2: **Left**: During the learning stage, action usage by agent and action optimization are repeatedly executed. The improved action is evaluated on the training instances, identifying the failed case for the subsequent learning step. Actions are improved through either updating the functions or writing notes. Multiple samples are produced during the learning, and upon evaluation against training instances, the optimal one is selected for the next iteration. **Right**: Case example of action updating and note writing. The action update addresses previous shortcomings by refining functions for improved issue resolution. Conversely, note writing advises agents on proper action usage. LLMs have the freedom to select from two learning options.

(3) `ActionLearn`: We address errors by implementing either function updates or writing notes. Function updates refines the Python function to correct the misunderstanding and overlooking of tasks, whereas writing notes entails enhancing the function's description to guide the agent toward more accurate use. The two options are both available and LLM *F* is free to choose one of them, as shown in Figure 2. When function updates are made, the corresponding action instructions $\pi_{\mathcal{A}_i}$ are also updated later to adapt to the new actions. Although function updating is triggered for a specific action function, it is not restricted to modifying just this single action. By accessing all action functions via prompt, LLM could choose by itself the actions to revise and could also incorporate new actions. Consequently, `ActionLearn` can freely update the entire action space in each iteration. The detailed prompt is in the Appendix.

**3. Augment Action Learning with Sampling** In practice, the action learner could generate low-quality actions that can greatly affect the efficiency of the overall optimization process. Also, some action revisions may be spurious. Thus to enhance the stability of the learning process and improve the quality of each step, we sample $K$ times with `ActionLearn`, yielding K revised results $\{\mathcal{A}_i^k, \pi_{\mathcal{A}_i}^k\}_{k=1,\ldots,K}$. During the learning iteration, each action-policy pair is evaluated on the training set, and the best one is chosen, as shown in Figure 2 (left) and the detailed algorithm in Appendix.

We select the best action based on the results obtained with the agent. The score $\mu$ for a sample $\mathcal{A}_i^k, \pi_{\mathcal{A}_i}^k$ is as follows:

$$\mu = p_{\text{succ}} + p_{\text{stepacc}}, \tag{3}$$

Here, $p_{\text{succ}}$ denotes the success rate of the agent in training instances, and $p_{\text{stepacc}}$ refers to the ratio of successfully executed actions to the total number of steps taken by the agent, indicating the practicality of generated actions. Based on these selection criteria, we identify the most beneficial and feasible action sample $\mathcal{A}_i, \pi_{\mathcal{A}_i}$.

### 4.2 Testing Stage

After completing the learning phase, the agent possesses an updated action space and refined policy instructions. In the testing phase, the agent attempts to solve problems in the same procedure as the `SolveProblem` process used during training. Notably, whereas

most prior research (Sun et al., 2023; Shinn et al., 2023) has concentrated on the learning loop within a single problem instance, LearnAct is designed to facilitate the transfer to this general type.

## 5 Experiment

### 5.1 Tasks

**Robotic Planning** (Ma et al., 2024) includes four challenging Robotic Planning tasks, namely Gripper, Blockworld, Barman, and Tyreworld. We followed the environment implementation of AgentBoard (Ma et al., 2024) and LLM+P (Liu et al., 2023). These tasks involve long-horizon robot planning problems, such as creating cocktails based on customer orders using available ingredients and containers, moving objects between different rooms using grippers, or rearranging piles of blocks to achieve a specified target configuration. These tasks put a significant demand on the agent's long-term planning capabilities.

**AlfWorld** (Shridhar et al., 2020) simulates six types of objectives within a household. For instance, agents are required to locate an apple within the house, heat it, and then place it in a target area. These tasks push the agent to explore the house systematically, given that the complete state of the environment is unknown to the agent.

### 5.2 Baselines

**Act & ReAct** (Yao et al., 2023a): The basic agent, Act, is prompted to iteratively take actions and obtain observations in the environment. Based on Act, the ReAct agent employs "Think" as an additional action.

**Reflexion** (Shinn et al., 2023): Reflexion is a learning method for language agents that utilize language as a learned policy. The original Reflexion is designed for learning individual policies in each instance. In our study, we adapt Reflexion for the training-testing setting, where policies are designed to be not instance-specific, facilitating transferability. Details are in the Appendix.

**CodeAsPolicy** (Liang et al., 2023): The method utilizes code to generate the entire solution for the task in an open-loop way. As the method does not interact with environments, we only report the results of CodeAsPolicy on Robotic policy tasks, because the ALfWorld task necessitates exploration in the environment to obtain information, and CodeAsPolicy cannot generate a code solution without access to the complete environmental information.

**Voyager** (Wang et al., 2023a): Voyager proposes functions as actions to solve tasks in a closed-loop agent manner. The original Voyager is specifically designed for Minedojo (Fan et al., 2022), emphasizing skill acquisition through a structured hierarchical task curriculum. For comparison, we reimplement it as Wong et al. (2023), which creates skills via code with basic verification.

### 5.3 Setting

We test on both GPT-4 and GPT-3.5 Turbo models as the LLM during testing. We employ GPT-4 throughout our learning for action creation and improvement, as well as the learning of Reflexion and Voyager. We set the temperature as 0.0 for consistency and reproducibility in our experiments. The sampling number in our method is set to 4. For each task, we randomly select 3 instances for the training set, with the remaining instances used for testing. We report test set results for our LearnAct and all baseline models. The primary evaluation metric was the task success rate. Each experiment is conducted three times per task, and we present the average results. We ensure a fair comparison by providing a single in-context example to LearnAct and all baselines. CodeAsPolicy and Voyager's action creation utilize the identical code example as ours.

|  | GPT-3.5 | | | | | GPT-4 | | | | |
|--|blockworld|gripper|barman|tyreworld|Avg.|blockworld|gripper|barman|tyreworld|Avg.|
|Act|0.0|5.9|0.0|10.0|4.0|57.1|58.7|64.7|62.1|60.7|
|ReAct|0.0|5.9|9.8|0.0|3.9|71.4|45.1|70.5|66.4|63.3|
|Act+Reflexion|0.0|11.8|4.0|0.0|3.9|62.1|58.8|70.6|37.1|57.2|
|ReAct+Reflexion|4.3|9.9|11.8|4.3|7.6|72.2|57.1|74.4|63.1|66.7|
|CodeAsPolicy|0.0|0.0|1.9|5.0|1.7|23.6|41.2|17.6|29.3|27.9|
|Voyager|18.6|2.0|6.0|19.3|11.5|66.4|76.5|80.5|65.0|72.1|
|LearnAct|10.0|34.8|15.0|32.9|**23.2**|73.9|82.5|87.4|87.6|**82.8**|

Table 2: Performance of LearnAct and baselines on Robotic Planning tasks.

|  | GPT-3.5 | | | | | | | GPT-4 | | | | | | |
|--|Put|Clean|Heat|Cool|Look|Put 2|Avg.|Put|Clean|Heat|Cool|Look|Put 2|Avg.|
|Act|34.7|15.1|13.0|14.2|48.1|23.6|24.8|77.7|50.5|13.0|42.8|40.8|66.6|48.6|
|ReAct|25.4|8.3|3.2|13.1|11.1|36.1|16.2|73.9|61.9|30.5|46.4|59.7|49.8|53.7|
|Act+Reflexion|44.5|22.6|24.9|26.2|33.6|11.8|27.3|71.4|47.5|6.8|35.1|80.2|52.4|48.9|
|ReAct+Reflexion|41.4|11.9|3.2|7.1|8.9|39.3|18.6|61.8|54.0|39.9|62.9|57.6|52.3|54.7|
|CodeAsPolicy|-|-|-|-|-|-|-|-|-|-|-|-|-|-|
|Voyager|31.5|27.4|11.7|7.2|22.6|9.6|18.3|84.1|60.8|27.9|49.7|68.6|57.1|58.0|
|LearnAct|65.1|36.1|30.3|42.0|24.1|16.9|**35.8**|85.6|76.2|53.2|72.0|64.5|81.5|**72.2**|

Table 3: Performance of LearnAct and baselines on Alfworld tasks.

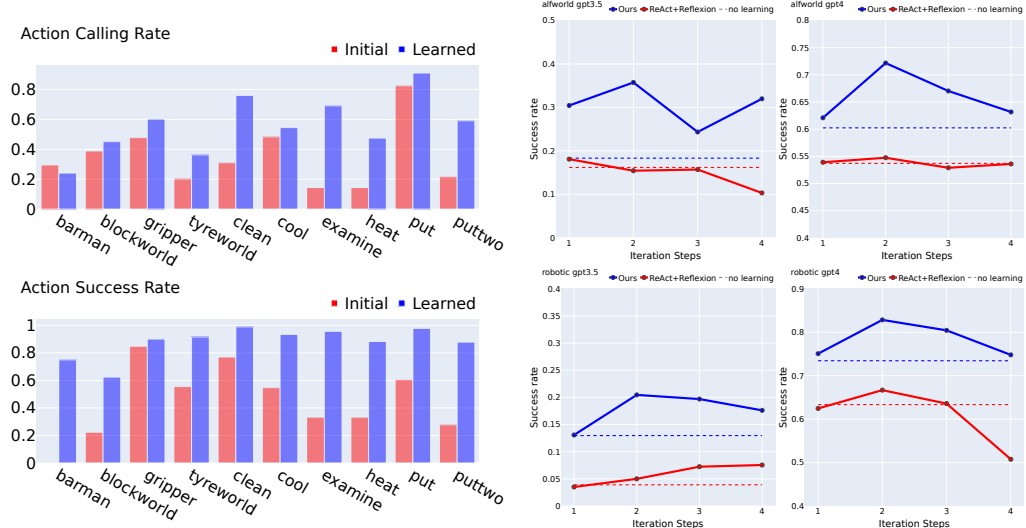

Figure 3: **Left**: The frequency of use and accuracy of learned actions before and after learning. Post-learning, there is a marked increase in the action usage frequency and accuracy, indicating their enhanced reliability and utility for the agent. **Right**: The performance with different maximum learning iteration steps. The performance varies with different maximum learning iteration steps. LearnAct's performance notably improves with the application of learning, particularly at step two. Although ReAct+Reflexion also shows improvement, its progress is less significant and stable.

## 5.4 Main Results

We present a comparison of LearnAct with baseline models in Tables 2 and 3. LearnAct outperforms the baselines in various tasks, utilizing both GPT-3.5 Turbo and GPT-4 as backbones. We further analyze the performance and discuss the conclusions drawn from the comparison.

**A well-designed action space enhances the language agent's ability to plan and solve tasks.** LearnAct markedly surpasses the Act baseline in performance, highlighting the crucial role of the action space. It is noteworthy that, after its learning phase, LearnAct is also an Act type agent except with a developed action space. This supports our hypothesis that the limitation of language agents lies in the effective integration of the planning process with the action space, and an adaptable action space can substantially unlock the latent planning capabilities of LLM. Additionally, LearnAct surpasses ReAct in both Robotic

Planning and AlfWorld, demonstrating that while "Think" improves planning, it suffers from ill alignment with the action space.

**Action learning outperforms the verbal policy learning.** In the realms of Robotic Planning and AlfWorld, LearnAct markedly outperforms both Act+Reflexion and ReAct+Reflexion. Reflexion writes policy prompts during the learning process to assist the agent in task-solving. This approach does yield improvements over the Act and ReAct methods; however, these enhancements are considerably less pronounced than those achieved by LearnAct, which focuses on learning within the action space. While learned verbal policies and prompts can aid in enhancing the agent's planning capabilities, possessing proficient skills is essential for the agent to fully realize its potential.

**Action learning is important for the action quality.** LearnAct outperforms coding baselines such as CodeAsPolicy and Voyager. CodeAsPolicy directly generates code in open loop without interacting with the environment, leading to inferior performance compared to closed-loop agent baselines. Notably, while Voyager generates actions using a method similar to LearnAct, it lacks the component of iterative action learning. This highlights the significance of action learning. Without this learning phase, the created actions are constrained by insufficient understanding of the task.

## 5.5 Analysis of the Learning Process

**Learning enhances performance through action reliability and utility.** We further analyze action utilization to explain why action learning enhances agent performance. We assess the frequency of use and accuracy at both the initial stage of learning and the post-learning. These results are presented in Figure 3 (left). It is evident that, compared to the initial action values, the actions post-learning demonstrate significantly higher correctness, indicating their enhanced reliability and utility for the agent. This is a direct consequence of our learning method, which focuses on correcting errors encountered during usage. Additionally, it is observable that the rate of action usage also improves after learning. It is possible because the agent is more inclined to employ these actions upon recognizing their usefulness, i.e., observing valid effects after calling the action.

| | Robotic Planning | Alfworld |
|---|---|---|
| Function Updating | 0.86 | 0.92 |
| Note Writing | 0.14 | 0.08 |

| | Robotic Planning | AlfWorld |
|---|---|---|
| Initial | 3.75 | 3.44 |
| After Learning | 3.83 | 3.50 |

Table 4: The ratio of two learning choices.  Table 5: Average number of learned actions.

**LearnAct learns action by updating functions, writing notes, and generating new actions.** We analyze the learning choices and learned action numbers. The strategies for learning encompass function updates and writing notes. We find that the model prefers to update functions. Table 4 shows function updating occurs for about 90 % in the learning. This may be due to the model's ability to circumvent most invalid executions through code adjustments. Additionally, the learning process can lead to the creation of new functions. As shown in Table 5, the average number of learned actions ranges from 3 to 4 for both Robotic Planning and AlfWorld, and the number of actions tends to increase after learning, showing that new actions could be innovated while updating existing ones.

**The iteration number influences the performance of the model.** We illustrate the impact of learning iterations on LearnAct and ReAct+Reflexion in Figure 3 (right). It is evident that in both AlfWorld and Robotic Planning tasks, LearnAct's learning markedly enhances performance, achieving optimal results within just two iterations. The enhancement from our learning approach is notably more substantial than that observed with Reflexion. While ReAct+Reflexion also shows performance gains, its impact is less consistent and displays more negative effects.

Why does prolonged learning with LearnAct detrimentally affect performance? We observe that as it cannot be avoided for agents to make mistakes, excessive optimization for these mistakes can lead to overfitting to specific training cases and misunderstanding the task rule. See failed cases in Appendix.

## 5.6 Ablation Study

We conducted ablation experiments to demonstrate the significance of various components in our method. First, we evaluated the action format of our method, including descriptions and usage examples of actions. The results of employing GPT-4 as the agent in Robotic Planning and Alfworld are presented in Table 6. It was observed that both the description and usage examples contribute to the performance, with the usage examples having a more pronounced impact compared to the descriptions. Interestingly, we discovered that the usage examples are often incorrect. Despite this, they still encourage the agent to utilize the learned actions. The descriptive aspect also plays a crucial role in Robotic Planning tasks. This could be attributed to that the actions have many pre-conditions, so the descriptions of the learned actions aid the agent in their correct usage.

|  | Robotic Planning | Alfworld |
|---|---|---|
| LearnAct | 82.8 | 72.2 |
| w.o. description | 78.5 | 72.4 |
| w.o. usage example | 77.4 | 65.4 |

Table 6: The ablation study on the form of actions, specifically the action description and usage examples of actions.

|  | Robotic Planning | Alfworld |
|---|---|---|
| LearnAct | 82.8 | 72.2 |
| sampling only | 73.4 | 60.2 |
| w.o. updating function | 75.3 | 62.0 |
| w.o. writing notes | 80.8 | 71.5 |
| w.o. sampling | 73.2 | 62.4 |

Table 7: The ablation study of the learning method, including the options during learning and the sampling method.

Next, we examine the components of our learning algorithm, which includes two types of learning: updating function and writing notes, as well as sampling during the learning process. The results are presented in Table 7. Function updating plays a crucial role in action learning, and writing notes for actions further enhances performance. We observed that function updating occurs approximately 90% times during learning, suggesting that the learner predominantly opts to modify functions to improve actions rather than merely adjusting notations for more accurate action use.

Moreover, sampling is vital for the effectiveness of the learning process. In the absence of sampling, the learner generates only a single action proposal for the next iteration, which is highly prone to errors and can negatively impact overall performance. However, note that sampling alone (w.o. learning) is ineffective in learning actions as LearnAct does, highlighting the importance of our learning algorithm.

## 6 Conclusion

In conclusion, our research advances LLM agents by equipping them with the ability to learn and refine actions through direct interaction with the environment. Our proposed framework LearnAct demonstrates a significant improvement in agent performance by enabling open-action learning, which aligns closely with how humans acquire and enhance skills. The empirical success of our methods in Robotic Planning and Alfworld environments underscores the potential of action learning in developing more intelligent and capable LLM agents.

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

# A Appendix

## A.1 The Action Learning with Sampling

Algorithm 1 illustrates the general framework for the learning iteration. In practice, we include multi-sampling in each learning iteration, to avoid the misdirection of low-quality solutions to the whole learning. Here we give a detailed algorithm of our learning method with sampling in Algorithm 2.

---

**Algorithm 2** Training Process of LearnAct with Sampling

---

**Input:** Training set $\mathbb{D}_{train} = \{\mathcal{E}_1, \ldots, \mathcal{E}_M\}$, LLM $G$.
**Input:** Original Actions $\mathcal{A}_0$, Basic Task Instruction $\pi_0$
**Input:** Sampling Number $K$
*# Training Procedure*
$\mathcal{A}_i^k, \pi_{\mathcal{A}_i}^k = \texttt{ActionCreation}(\pi_0), k = 1 \textbf{ to } K$
**for** $i = 1$ **to** maxiter **do**
    results$^k$ = $\texttt{SolveProblem}(\pi_0 + \pi_{\mathcal{A}_i}^k, \mathcal{E}_m, \mathcal{A}_0 \cup \mathcal{A}_i^k, G), m = 1 \textbf{ to } M, k = 1 \textbf{ to } K$
    *# Compute score and select the best sample*
    $\mu^k = p_{\text{succ}}^k + p_{\text{stepacc}}^k, k = 1 \textbf{ to } K$
    $k_{\text{best}} = \arg\max_k \mu^k$
    results, $\pi_{\mathcal{A}_i}, \mathcal{A}_i = $ results$^{k_{\text{best}}}, \pi_{\mathcal{A}_i}^{k_{\text{best}}}, \mathcal{A}_i^{k_{\text{best}}}$
    $s_{1:T}, a_{1:T}, r = \texttt{SelectErrorCase}(\text{results})$
    $\mathcal{A}_{i+1}^k, \pi_{\mathcal{A}_{i+1}}^k = \texttt{ActionLearn}(\mathcal{A}_i, s_{1:T}, a_{1:T}, r), k = 1 \textbf{ to } K$
**end for**
**Return** $A_{\text{last}}, \pi_{A_{\text{last}}}$

---

## A.2 Complexity Analysis

The testing stage of LearnAct follows the standard language agent setting, except that the action space is augmented with manufactory actions. The overall time complexity is thus the same as the standard Act method. As for the learning stage, the time complexity is $O(MKI)$, where $M$ is the training instance number, $K$ is the sampling number, and $I$ denotes the max iteration number.

## A.3 A Bayesian View of Open-Action Learning

In this subsection, we present a Bayesian perspective on our learning approach. We express the transition function $\mathcal{T}$ and reward function $\mathcal{R}$ in probabilistic terms $p(s_{t+1}|s_t, a_t)$ and $p(r|s_{1:T})$. Denote the observation as function of state $o_t = \text{o}(s_t)$, and the history at time $t$ is denoted as $\text{h}(s_{1:t}, a_{1:t-1}) = [\text{o}(s_1), a_1, \ldots, \text{o}(s_{t-1}), a_{t-1}, \text{o}(s_t)]$, i.e., the function of $s_{1:t}, a_{1:t-1}$. The distribution of a Partially Observable Markov Decision Process trace is as follows:

$$p_{\mathcal{A}, \pi_{\mathcal{A}}}(s_{1:T}, a_{1:T}) = p(s_1) \prod_{t=1}^{T-1} [\pi_{\mathcal{A}}(a_t|\text{h}(s_{1:t}, a_{1:t-1})) p(s_{t+1}|s_t, a_t)] \pi_{\mathcal{A}}(a_T|\text{h}(s_{1:T}, a_{1:T-1})) \quad (4)$$

The reward $r$ conditioned on action space $\mathcal{A}$ and policy $\pi_{\mathcal{A}}$ follows the distribution:

$$p(r|\mathcal{A}, \pi_{\mathcal{A}}) = \mathbb{E}_{p_{\mathcal{A}, \pi_{\mathcal{A}}}(s_{1:T}, a_{1:T})} p(r|s_{1:T}) \quad (5)$$

The objective of POMDP is to maximize the reward, which can be seen as the posterior of action space $\mathcal{A}$ and policy $\pi_{\mathcal{A}}$ given the maximum reward:

$$p(\mathcal{A}, \pi_{\mathcal{A}}|r = r_{max}) = \frac{p(\mathcal{A}, \pi_{\mathcal{A}}) \mathbb{E}_{p_{\mathcal{A}, \pi_{\mathcal{A}}}(s_{1:T}, a_{1:T})} p(r = r_{max}|s_{1:T})}{Z_{r_{max}}} \quad (6)$$

where $Z_{r_{max}} = \mathbb{E}_{p(\mathcal{A},\pi_{\mathcal{A}})}\mathbb{E}_{p_{\mathcal{A},\pi_{\mathcal{A}}}(s_{1:T},a_{1:T})}p(r = r_{max}|s_{1:T})$ is the normalization factor. In conventional reinforcement learning, $\mathcal{A}$ is predetermined and remains constant, while the model only learns $\pi_{\mathcal{A}}$ through maximum likelihood rather than posterior estimation. This is attributed to the complexity of computing the posterior, which is due to the necessity of domain knowledge for the prior $p(\mathcal{A},\pi_{\mathcal{A}})$ and the intractable calculation of the posterior.

However, leveraging the advanced capabilities of language models, we propose employing them to estimate the posterior of the action space and policy toward the open action space learning. The prior $p(\mathcal{A},\pi_{\mathcal{A}})$ can be defined as the operator ActionCreation, and the posterior updating can be inferred as ActionLearn.

### A.4 Detailed Prompt

We present the prompt used in our experiment here, due to space limitations of the main text.

We first present the prompts for action learning. The ActionCreation process entails two phases: initially, function generation is guided by Prompt 4, followed by the crafting of function descriptions and usage instructions, as directed by Prompt 5 and 6, respectively.

During the action learning iterations, ActionLearn leverages Prompt 7 to derive results. If the LLM suggests updates to action functions, descriptions and usage instructions are regenerated using the Prompt 5 and 6.

---

**Prompt for Action Creation**

{basic task instruction}

Please propose several high-level steps for this task.

Each high-level step should be a Python function encompassing multiple (at least two) basic actions. All the values used in the function should be given as input rather than fixed in the function.

The provided actions are Python functions and can be executed directly, for example,
```python
{basic action example}
`

No additional interfaces besides the provided actions are available. All the code should be wrapped by```python
```

Here are examples: {created action example}

Now please write your solution:

---

Figure 4: Prompt for action creation.

---

**Prompt for Action Description Generation**

{basic task instruction}

Now here are some Python function encompassing multiple basic actions to serve as high-level interface in this task. Please write interface instruction for the given Python function.

---

Here are examples:

{action description examples}

Now please write interface instruction for this high-level step {func name}:
Function:
```python
{function}
```

Instruction:

Figure 5: Prompt for action description generation.

---

**Prompt for Action Usage Example Generation**

{basic task instruction}

Now, here are some Python functions encompassing multiple basic actions to serve as high-level interfaces in this task. Please complete the task with the interface following the format of the examples. {format instruction}

Here are examples:

{action usage examples}

Now, please complete the task using these interfaces:
Function:
```python
{function}
```Example:

Figure 6: Prompt for action usage example generation.

---

**Prompt for Action Learning**

{basic task instruction}
{task goal}

The actions provided are Python functions and can be executed directly, for example,
```python
{action example}
```

No additional actions besides the provided actions are available. All the code should be wrapped by```python
```

Now here are some high-level steps to complete this task. Each high-level step is a general Python function encompassing multiple (at least two) basic actions. All the values used in the function should be given as input rather than fixed in the function.

The high-level steps are executed but failed. Please analyze why the execution failed, and give one of the following improvement: Update, Plan. Please respond in the following format:
Failed reason: <>
Improve: <Update or Plan: [The target function]>
Content: <>
Test case: <>(This is only for Update case, not for Plan)

Here are examples:

{in context example}

Now please analyze this case:
```python
{actions}
```

The agent performs this task, and the high-level action {function name} is executed last:
{agent trajectory}

But an error is observed in the last call ({error info}). The detailed subprocess of this step is:
{error subprocess}

Failed reason:

Figure 7: Prompt for action learning.

We then show the prompts during the testing stage, i.e. the prompt of our agent. It is the same as the basic Act agent, as shown in the Prompt 8 for the Blockworld task.

**Prompt Details for Blockworld**

**System Prompt**
You are a master in planning.

- - - - - - - - - - - - - - - - - - - - - - - - - - - - - - - - - - - - - - - - - - - - - - - - -

**Instruction**
The domain assumes a world where there are a set of blocks that can be stacked on top of each other, an arm that can hold one block at a time, and a table where blocks can be placed.
The actions defined in this domain include:
Pickup(block): allows the arm to pick up a block from the table if it is clear and the arm is empty. After the pickup action, the arm will be holding the block, and the block will no longer be on the table or clear.
Putdown(block): allows the arm to put down a block on the table if it is holding a block. After the putdown action, the arm will be empty, and the block will be on the table and clear.
Stack(block1,block2): allows the arm to stack a block on top of another block if the arm is holding the top block and the bottom block is clear. After the stack action, the arm will be empty, the top block will be on top of the bottom block, and the bottom block will no longer be clear.
Unstack(block1,block2): allows the arm to unstack a block from on top of another block if the arm is empty and the top block is clear. After the unstack action, the arm will be holding the top block, the top block will no longer be on top of the bottom block, and the bottom block will be clear.

{learned action instructions}

Here are examples:
Goal: The goal is to satisfy the following conditions: b1 is on b2., b2 is on b3.
Observation: b2 is on b3. b3 is on b1. b1 is on the table. Robot arm is empty. The b2 is clear.
Action: Unstack('b2','b3')
Observation: b3 is on b1. b1 is on the table. The b3 is clear. You are holding b2.
Action: Putdown('b2')
Observation: b3 is on b1. b1 is on the table. b2 is on the table. Robot arm is empty. The b2 is clear.
Action: Unstack('b3','b1')
Observation: b1 is on the table. b2 is on the table. The b1 is clear. The b2 is clear. Robot arm is empty. You are holding b3.
Action: Putdown('b3')
Observation: b1 is on the table. b2 is on the table. b3 is on the table. Robot arm is empty. The b1 is clear. The b2 is clear. The b3 is clear.
Action: Pickup('b2')
Observation: b1 is on the table. b2 is on the table. The b1 is clear. The b3 is clear. You are holding b2.
Action: Stack('b2','b3')
Observation: b1 is on the table. b2 is on b3. b3 is on the table. Robot arm is empty. The b1 is clear. The b2 is clear.
Action: Pickup('b3')
Observation: The action is not valid and therefore takes no effect. Please remember to satisfy the restriction of actions.
Action: Pickup('b1')
Observation: b2 is on b3. b3 is on the table. The b2 is clear. You are holding b1.
Action: Stack('b1','b2')
Observation: b1 is on b2. b2 is on b3. b3 is on the table. Robot arm is empty. The b1 is clear. The goal is satisfied.

{usage example for learned actions}

{goal of testing task}
{history trajectory}
Action:

Figure 8: Prompt details of agent testing for task Blockworld.

## A.5 Learned Result Case

We illustrate the learned actions and corresponding instructions in Figure 9. The {learned action instructions} and {usage example for learned actions} are used in Prompt 8 to inform the agent model about how to use the learned action function.

Learned Result Case for Blockworld

**learned action function**

```
def dismantle_stack_until(block_list, block_target):
    for top_block, bottom_block in zip(block_list, block_list[1:]):
        if top_block == block_target:
            break
        Unstack(top_block, bottom_block)
        Putdown(top_block)

```

```
 8
 9  def construct_stack(block_list):
10      for top_block, bottom_block in reversed(list(zip(block_list,
        block_list[1:]))):
11          Pickup(top_block)
12          Stack(top_block, bottom_block)
```

----------------------------------------------------------

**learned action instructions**
dismantle_stack_until(block_list, block_target): Allows the arm to dismantle a stack of blocks one by one, stopping when it reaches a specific target block. 'dismantle_stack_until(block_list, block_target)' sequentially unstacks each block from the block list starting from the top and places it on the table until it reaches the target block. The blocks in the list must be clear and stacked consecutively. If the top block from the list matches the target block, the function ends, leaving the arm empty and the blocks dismantled on the table. For example, if blocks b3, b2, and b1 are clear with b3 on top of b2 and b2 on top of b1, and the arm is empty, 'dismantle_stack_until(['b3','b2','b1'], 'b2')' will unstack b3 from b2 and put b3 on the table without touching b2 or b1.
construct_stack(block_list): This function allows the arm to construct a stack of blocks given a list of blocks if the arm is empty and all the blocks are clear and on the table. It achieves this by iterating from the end of the list, picking each block starting from the penultimate block and stacking it on the block next to it (assuming the list is arranged from bottom block to top block). After the execution of construct_stack, all the blocks in the list will be stacked on top of each other in the order they were arranged in the block_list, the arm will be empty and only the top block will be clear. For example, if block1, block2, and block3 are all clear and on the table and the arm is empty, calling construct_stack([block1, block2, block3]) will result in block3 being stacked on block2 and block2 on block1, and the arm will be empty.

----------------------------------------------------------

**usage example for learned actions**
The goal is to satisfy the following conditions: b1 is on b2, b2 is on b3.
Observation: b3 is on b2, b2 is on b1, b1 is on the table. Robot arm is empty. The b3 is clear.
Action: dismantle_stack_until(['b3','b2','b1'], 'b1')
Observation: b1 is on the table. b2 is on the table. b3 is on the table. Robot arm is empty. The b1 is clear. The b2 is clear. The b3 is clear.
Action: construct_stack(['b1','b2','b3'])
Observation: b1 is on b2. b2 is on b3, b3 is on the table. Robot arm is empty. The b1 is clear. The goal is satisfied.

Figure 9: Prompt details of agent testing for task Blockworld.

## A.6 Experiment Details

### A.6.1 Baselines

We imply baseline Reflexion for our learning-testing setting, where the testing instances are unseen during learning. Original Reflexion learns policy on a single instance, which can generate very detailed hints for the next turn, like "In this trial, I was able to find the desklamp on desk 1, but I did not find the bowl under the desklamp. In the next trial, I will go to desk 1, turn on the desklamp, then look for the bowl under the desklamp on desk 1 or desk 2. If I still cannot find the bowl, I will check the shelves and drawers." The detailed hint helps the agent perform in the same task instance, but can not transfer to different instances. We imply Reflexion to generate transferable policies based on the failed trial, such as "To accomplish the task, first locate the object, then navigate towards the cleaning location, clean the object, and finally deposit it at the specified receptacle. Follow the action

```
def find_and_take(obj, recep):
    observation = goto(recep)
    object_name = parse_object_name(obj, observation)
    if object_name is not None:
        observation = take(object_name, recep)
    return observation
```

To find the object I need, I should explore
different receptacles.

Error: The action is not valid. The cabinet 2 is closed.

Failed Trail. In the trial, saw a closed cabinet but do not open it.

Update:

Reflexion:

```
def find_and_take(obj, recep):
    observation = goto(recep)
    if not any([recep.startswith(location) for
location in ['shelf', 'countertop']]):
        observation = open(recep)
    if 'is open' in observation:
        object_name = parse_object_name(obj,
observation)
        if object_name is not None:
            observation = take(object_name, recep)
    return observation
```

Before attempting to interact with the
receptacle, ensure that it is open or
accessible in the first place. Validate
object's existence and accessibility in the
said location.

Figure 10: The failed cases of LearnAct (left) and Reflexion (right). The learning overfits the current task instance with misunderstanding. Left: The agent causes error when using action find_and_take because the action does not consider the case that the cabinet is closed. However, the learner updates action find_and_take by adding a conditional based on the receptacle name, which is not the correct condition to open the receptacle. Right: The agent failed in a trial in which it saw a closed cabinet but did not open it. The Reflexion learner advises the misleading policy "before attempting to interact with the receptacle, ensure that it is open or accessible in the first place".

sequence: locate - navigate - clean - store. Ensure to recheck your inventory to confirm the success of your actions.". The experiment shows that general policy can improve agent performance, as shown in Table 2 and 3.

### A.6.2 Setting

We use language models GPT-4 and GPT-3.5 Turbo through interface openai.ChatCompletion.create via Azure platform. For all the learnable models, including LearnAct, Reflexion, and Voyager, GPT-4 serves as the learner for both GPT-4 and GPT-3.5 Turbo agents. This choice is driven by the necessity for substantial model capacity to facilitate agent learning, aligning with original papers of baselines where learning is implied by strong language models.

### A.6.3 Case Study for Failed Learning

Figure 3 demonstrates that excessive iterations can degrade performance, likely due to overfitting and misinterpretations of the learning task. Failed cases involving LearnAct and Reflexion are shown in Figure 10, in which the learner inaccurately diagnoses failure causes, resulting in misguided actions or policies that further impair performance.

