# OpenReview forum: "Empowering Large Language Model Agents through Action Learning"
_colmweb.org/COLM/2024/Conference — COLM_

### Official Review · Reviewer_n7eb · 2024-05-02

**Rating:** 6
**Confidence:** 4
**Ethics Flag:** 2

**Summary:**

This research paper introduces an innovative framework named LearnAct, designed to enhance the capabilities of LLM agents through action learning. The study tackles the limitations of fixed action spaces in LLM agents by allowing these agents to learn and refine their actions based on trial and error, a process akin to human experiential learning. This is achieved through an iterative learning strategy where actions are continuously updated based on the outcomes of previous tasks. The authors demonstrate the effectiveness of this approach in the Robotic Planning and Alfworld environments, showing significant improvements in task performance.

**Ethics Concerns Details:**

The newly-generated actions may require ethics check

**Questions To Authors:**

1. **Comparative Analysis:** Could the authors provide a more detailed comparative analysis with other methods in action learning for LLM agents? Specifically, how does LearnAct perform compared to methods that do not use iterative learning strategies?
2. **Baseline Selection:** The authors claimed that LearnAct achieves more effective and efficient learning compared to state-of-the-art methods, but the baselines are not state-of-the-art and up-to-date enough. In fact LearnAct performs worse than some up-to-date models, such as FireAct [6] and KnowAgent [7].
3. **Flexibility:** Is it necessary to “Expanding Action Space by Action Creation” first? If not, how can you adaptively decide to expand the action space or not? If so, could you explain why action space expansion is a necessity, as the current action space is sufficient enough in many cases.
4. **Generalizability:** Can the authors elaborate on how the LearnAct framework can be adapted for use in environments or tasks significantly different from those tested (e.g., non-robotic applications)?
5. **Scalability Concerns:** How does the LearnAct framework scale with increasingly complex tasks or larger environments? Are there computational constraints that could limit its practical applicability?
6. **Overfitting Measures:** What measures have been implemented in the LearnAct framework to prevent overfitting during the iterative learning process? How do you ensure that the actions learned are not overly tailored to the specific training examples used?

**Reasons To Accept:**

1. **Innovative Approach:** The concept of open-action learning to dynamically expand and refine actions for LLM agents is novel and addresses a critical limitation in the current use of LLM agents in various applications.
2. **Thorough Experimentation:** The authors provide extensive experimental results to validate their claims. The use of multiple environments for testing and the iterative approach to learning and refining actions are well-executed and robustly documented. Although the baseline selection is not sound enough.
3. **Potential for Broader Implications:** The study’s findings could have significant implications for the development of more autonomous and efficient LLM agents across a wide range of applications, potentially leading to broader advancements in artificial intelligence.

**Reasons To Reject:**

1. **Lack of Comparative Analysis:** While the paper shows that LearnAct outperforms certain existing methods, there is a lack of comprehensive comparative analysis with a wider range of existing solutions (especially SOTA and more up-to-date methods), which could provide a better understanding of LearnAct’s positioning in the field.
2. **Complexity and Replicability:** The complexity of the LearnAct framework and its dependency on specific environments and configurations may limit its replicability and broader application. The paper could benefit from a clearer explanation of how LearnAct can be adapted or generalized for other contexts.
3. **Potential Overfitting:** There is a concern about the potential for overfitting in the iterative learning process, especially given the significant improvements reported. The paper could be strengthened by including a discussion on measures taken to avoid overfitting and ensure the generalizability of the learned actions.

---

> ### Author Rebuttal · Authors · 2024-05-31
>
> Thank you for your questions and suggestions!
>
> **Comparative Analysis**: In Experiment, CodeAsPolicy and Voyager are baselines without iterative learning strategies. The Ablation Study also shows LearnAct's performance without iterative learning.
>
>
> **Baseline Selection**: Thanks for the baseline suggestion! FireAct and KnowAgent are based on fine-tuning, distinguishing them from the prompting-based learning approach in this study, which may lead to better performance. We'll revise our statement and include these works in the related section.
>
>
> **Flexibility**: This is a good question! As outlined in the introduction, our hypothesis is that for most multi-turn agent tasks, abstracting higher-level actions is reasonable, consistent with hierarchical reinforcement learning's core assumption. Theoretically, frequent patterns in trajectory can be abstracted as higher-level skills, emphasizing their semantic meaning, especially for complex tasks with extended action sequences.
>
> Technically, besides semantic judgment, we can implement action learning and evaluate performance to determine if action space learning is necessary. Improved performance indicates the need for action space expansion; otherwise, the existing action space may be adequate, which we believe is rare.
>
>
> **Generalizability**: Our framework targets general multi-turn decision-making agents, with the agent's form Act being versatile for various tasks. The learning method is generalizable to many tasks like OS operation, Sheet operation, and tool usage scenarios where actions can be defined as code.
>
>
> **Scalability Concerns**: The Appendix shows LearnAct's testing phase time complexity adheres to the standard language agent setting, dependent on prompt size O(N), where N is the number of learned actions. The learning phase time complexity is O(MKI), with M being the number of training instances, K the sampling number, and I the maximum iteration step.
>
> LearnAct suits most current language agent task scales. For extremely complex tasks, a retriever-based skill selector might be needed due to the vast array of potential skills. We will add this discussion in the Appendix.
>
>
> **Overfitting Measures**: We mitigate overfitting by requiring action generation and updating instructions "All the values used in the function should be given as input rather than fixed in the function.", preventing overfitting to training instances. We also limit training iteration to avoid overfitting.

---

> > ### Comment · Reviewer_n7eb · 2024-06-04
> >
> > Dear Authors,
> >
> > Thank you very much for the clarification! Most of them have addressed my concerns.
> >
> > I appreciate the response, to include those in the revised draft will strengthen the paper.
> >
> > Overall, I acknowledge the idea of this work, but still have reservations about viewpoints of writing quality, and decide to increase my original scores 5 to 6.
> >
> > Best Regards,
> >
> > Reviewer n7eb

---

> > > ### Author Response · Authors · 2024-06-04
> > >
> > > We sincerely appreciate your feedback! We remain available to address any further questions you may have.

---

### Official Review · Reviewer_ADgy · 2024-05-08

**Rating:** 6
**Confidence:** 3
**Ethics Flag:** 1

**Summary:**

The authors present alba approach for LLM agents to learn new actions and thereby improve the ability to successfully perform tasks in a simulation. The presented idea to learn new actions formalized as python code and how to integrate this new ablities into the policy of the agent.
The approach is overall presented clearly, however some details could be explained in more detail.
The approach is very well evaluated on two different tasks (Robotic Planning and AlFWorld). Thereby, the authors compare the method to several related work and the presented results are promising.
The authors missed a detailed analyze of the influence of the LLM. The LLM is only treated as a black box. The influence of the quality of the LLM is only analyzed very briefly and the authors do not evaluate any open source LLM to perform a more detailed analyze

**Questions To Authors:**

Tabel 5: How can the model use learn actions before learning?

**Reasons To Accept:**

Good motivated approach to improve the performance of LLM Agents
Detailed analysis of the presented approach
Promising results

**Reasons To Reject:**

The influence of the LLM on the overall performance is only investigated very briefly.

---

> ### Author Rebuttal · Authors · 2024-05-31
>
> Thank you for your questions and suggestions!
>
> Influence of LLM: We perform experiments using Mistral-7b and DeepSeek-67b as backbone models. The results for Robotic Planning are as follows:
>
> |  | Mistral-7b | DeepSeek-67b | GPT-3.5 | GPT-4 |
> |---|---|---|---|---|
> | ReAct+Reflection | 3.5 | 8.7 | 7.6 | 66.7 |
> | LearnAct | 5.1 | 20.8 | 23.2 | 82.8 |
>
> For powerful open-source LLMs like DeepSeek-67b, LearnAct can enhance performance. For weaker models, task completion remains challenging. Note that action learning iteration for Mistral-7b is difficult as the trajectory offers limited information for the learner to update actions.
>
>
> Before learning in Table 5: Our apologies for any ambiguity. The terms "init" and "after learning" refer to the scenario in which actions and policies are initially generated through the ActionCreation process and subsequently undergo iterative refinement via ActionLearn. To enhance clarity, we will modify the column names to "init" and "after updating." Prior to the update, the actions function in an identical manner to their post-update counterparts, as ActionLearn merely serves to optimize the action and instruction. As depicted in Figure 1 (right) and Figure 8 in the Appendix, the instructions for the generated actions are provided to the Act agent similarly to predefined actions.

---

> > ### Author Response · Authors · 2024-06-04
> >
> > We sincerely hope that our response addresses your questions. We remain available to address any further questions you may have.

---

> > ### Comment · Reviewer_ADgy · 2024-06-05
> >
> > Thanks for the additional experiments with other LLMs. This information is very helpful. And also the question is answered well.

---

> > > ### Author Response · Authors · 2024-06-06
> > >
> > > Thank you for your valuable feedback! We are delighted to have addressed your questions and remain available to address any further questions you may have.

---

### Official Review · Reviewer_vBTB · 2024-05-10

**Rating:** 7
**Confidence:** 5
**Ethics Flag:** 1

**Summary:**

This paper is about improving the performance of language agents by refining the action space during the learning phase. Specifically, the authors propose LearnAct, an iterative system that uses LLMs to create new (or edit existing) actions based on past experiences and to act as a policy. In this work, the actions are expressed as Python code which enables the use of conditional statements, loops, and function calls (i.e., composition of actions). Experiments were conducted on robotic planning tasks and ALFWorld (a text-based game version of ALFRED, a 3D embodied environment). Empirically, the authors show that LearnAct achieves better performance on average compared to some relevant baselines (ReAct, Reflexion, CodeAsPolicy, Voyager) on those environments.

Edit: I have increased my score to 7.

**Questions To Authors:**

- What is the definition of closed-loop in the context of this work? Is it the action learning process that is closed-loop or the actions (i.e., Python functions) themselves? If you are referring to the actions, have you checked if they are really closed-loop? In my experience Voyager tends to generate actions that are open-loop, i.e., they ignore the state of the environment as they send command to the environment. Granted this might be less problematic in your environments compared to Minecraft where other entities are moving in parallel.
- In section 5.3, it says the **temperature** is 0, but it also mentions sampling number is 4. I'd expect to get very similar samples (if not all the same). Can the authors clarify this?
- Do the authors have any insights on the generalization of the learned actions to unseen domains but that are sharing the same predefined actions?
- I'm curious to know if the authors have considered any open-source models during their experimentation?

**Reasons To Accept:**

- Letting the the action space evolve based on past experiences is a nice idea. This is a good way to get actions that operated at different level of abstraction, i.e. hierarchical planning.
- Having higher-level actions (expressed as code) help reducing the compute cost during inference. That is, you don't have to call the LLMs at each timestep.
- Transferability of the learned actions to new tasks from a same domain. I think this is a very important aspect of the work, as many prior works focuses on single-problem solving.

**Reasons To Reject:**

- The research on Language Agents is evolving very fast and there is probably several prior works missing from the related work section. On top of my head, two came in mind: [TaskWeaver: A Code-First Agent Framework](https://arxiv.org/abs/2311.17541) and [ReGAL: Refactoring Programs to Discover Generalizable Abstractions](https://arxiv.org/abs/2401.16467). I think it would be beneficial to the reader to have a more comprehensive related work section to better position their work.

- This is a smaller concern. I noticed a small paragraph in A.6.3 about failure mode of the proposed technique regarding excessive iterations (which according to Fig.3 anything above 2 iterations looks worst!). Is this that the only failure? I doubt it, e.g., limit of the context length in ActionLearn to update the full action space at each iteration. I think having a proper discussion of the limitation as part of the main text would strengthen the paper.

---

> ### Author Rebuttal · Authors · 2024-05-31
>
> Thank you for your questions and suggestions!
>
> Related work: In the next version, we'll include TaskWeaver, ReGAL, and other recent works.
>
> Limitations: Other potential factors include the learner's capacity, the constraint on the number of samples, and so forth. We'll add a section on limitations in the next version.
>
>  Closed-loop Definition: We use "closed-loop" to differentiate CodeAsPolicy from alternatives, as it generates the entire code without agent feedback processing. Our method focuses on multi-turn agents capable of accessing environmental feedback after executing a action (not mean the action to process feedback). Indeed, Voyager's original method does not constitute an Act-type agent, as it generates a one-time solution to the task while iterating on feedback to refine the code outcome. We will modify the term for clarity.
>
> Sampling Temperature: We conducted experiments with different sampling temperatures and found that a temperature of 0 still produces varied skill propositions due to GPT's randomness. The ablation study of sampling supports this. We present the temperature's impact as follows:
>
> |  | Robotic Planning | AlfWorld |
> |---|---|---|
> | t=0, K=4 | 82.8(5.2) | 72.2(4.4) |
> | t=0.7, K=4 | 78.9(7.8) | 72.7(7.4) |
> | t=0, K=8 | 82.6(5.5) | 73.5(4.1) |
> | t=0.7, K=8 | 83.1(5.8) | 73.9(6.5) |
>
> K is sampling number. The number in () represents the std. For K=4, higher temperatures don't significantly improve performance but increase variance. To capitalize on sampling variance, a larger sampling number is more advantageous. The results indicate that a greater sampling temperature necessitates an increased sampling size to attain improved and consistent performance.
>
> Domain Generalization: This is an interesting idea! Such a setting would be more common in areas like OS operation and Sheet Operation, where various tasks share a common fundamental interface.
>
> Open-source Models: We perform experiments using Mistral-7b and DeepSeek-67b as backbone models. The results for Robotic Planning are as follows:
>
> |  | Mistral-7b | DeepSeek-67b | GPT-3.5 | GPT-4 |
> |---|---|---|---|---|
> | ReAct+Reflection | 3.5 | 8.7 | 7.6 | 66.7 |
> | LearnAct | 5.1 | 20.8 | 23.2 | 82.8 |
>
> For powerful open-source LLMs like DeepSeek-67b, LearnAct can enhance performance. For weaker models, task completion remains challenging. Note that action learning iteration for Mistral-7b is difficult as the trajectory offers limited information for the learner to update actions.

---

> > ### Comment · Reviewer_vBTB · 2024-06-03
> >
> > Thank you for the insightful rebuttal. All my questions have been answered and the authors acknowledged that my *minor* concerns will be addressed in the final version. As far as I know evolving the action space based on gather experiences as proposed is novel (at least novel as it can be in this rapidly evolving research area). I will raise my score to 7 to help with potential tiebreak.

---

> > > ### Author Response · Authors · 2024-06-04
> > >
> > > We sincerely appreciate your valuable feedback! We remain available to address any further questions you may have.

---

### Official Review · Reviewer_X3uD · 2024-05-15

**Rating:** 4
**Confidence:** 4
**Ethics Flag:** 1

**Summary:**

This paper proposes a training framework that instructs LLMs to learn new actions. During the training stage, the action is created in the form of executable code function, and is refined through execution feedback. In the inference time, the LLM can utilize the created action to tackle more novel scenarios and perform more complex tasks.

**Questions To Authors:**

None

**Reasons To Accept:**

1. The idea is interesting, and can potentially solve the problem of constrained action space.
2. The experimental results demonstrate the effectiveness of the proposed framework.

**Reasons To Reject:**

Overall, the idea is intuitive and makes sense to me. But the biggest concern is that I do not find any fundamental difference and novel parts that distinguish this work from some previous publications, including but not limited to CRAFT, and Voyager. Please correct me if I have any misunderstanding. Basically, I think the fundamental part is creating some tools or actions defined in this work during the training time that are reusable, which can help LLMs to address some special test cases. Which part is novel in this work?

---

> ### Author Rebuttal · Authors · 2024-05-31
>
> Thank you for the question! In our work, we have performed a comparative analysis with Voyager in both related work and experiment. We will add CRAFT in the related work in the next version.
>
> Although function creation has attracted significant interest as a general technique recently, prior studies such as CRAFT and Voyager primarily emphasize coding-based task resolution, acquiring tools from the task prior and optimizing them via code refinement. These approaches are not applicable to interactive decision-making language agents. Our study differs from these works by targeting the action space learning of interactive decision-making language agents. LearnAct's learning strategy consistently refines the entire action space according to the task-solving trajectories, introducing an innovative and general method for action learning in interactive decision-making language agents.
>
> In one-turn coding tasks (LATM, CREATOR, and CRAFT), the function abstracted from one example can be easily used to tackle similar examples, Consequently, their learning approach is devised to optimize the performance of the single tool's code in the training instances through a code-refine iteration.
>
> Voyager suggests a curriculum for MineDojo that advances from elementary to complex levels within the hierarchical task framework. The solution for low-level tasks can naturally serve as a tool for higher-level tasks. The learning method also optimizes the code solution for the current task in a code-refine iteration, as Voyager resolves each task via open-loop code writing.
>
> None of these studies can be directly applied to general interactive decision-making agents. For such tasks, there is no predefined skill structure. In LearnAct, skills are generated through task decomposition based on the language model's commonsense. Moreover, our learning-testing framework addresses the iterative refinement of action space based on historical trajectories. This poses a considerable challenge, as the learner must determine why specific actions within the trajectory are suboptimal and improve them to support the agent's accomplishment of the overarching goal.
>
> We delineate the differences in a table:
>
> |  | Task | Tool Prior | Tool Learning |
> |---|---|---|---|
> | CRAFT,LATM | Coding | Given sample | Code revision |
> | Voyager | Coding | Given sample and Curriculum | Code revision |
> | LearnAct (Ours) | Interactive decision-making | None (Proposed by LM) | Trajactory based skill reflecion |

---

> > ### Author Response · Authors · 2024-06-04
> >
> > We sincerely hope that our response addresses your questions. We remain available to address any further questions you may have.

---

### Comment · Area_Chair_4oZK · 2024-06-03
**Discussion period**

Hi reviewers! This is a friendly reminder that the authors have posted responses to the initial set of reviews. I'd appreciate it if you could look through the authors' comments soon and let us know if your concerns have been addressed / if there are any outstanding issues soon so there's time to discuss.

With appreciation,
Your AC

---

### Decision · Program_Chairs · 2024-07-10

**Decision:**

Accept

**Comment:**

This paper describes a structured prompting method for using LMs to solve interactive tasks by writing (and iteratively refining) code fragments implementing high-level actions. It outperforms a variety of other prompting methods on two simulated interactive tasks. As with  ReAct, Voyager, and other work in this space, it's a little unclear whether there's a generalizable scientific finding here, or just a well-designed collection of prompts and some code gluing them together. But empirical results are good, the engaged reviewers appear enthusiastic, and so the paper seems likely to be a useful contribution in this sub-area.

[At least one review was discounted during the decision process due to quality]